# Mineral Nutrition of Naturally Growing Scots Pine and Norway Spruce under Limited Water Supply

**DOI:** 10.3390/plants11192652

**Published:** 2022-10-09

**Authors:** Yury V. Ivanov, Ilya E. Zlobin, Alexander V. Kartashov, Alexandra I. Ivanova, Valery P. Ivanov, Sergey I. Marchenko, Dmitry I. Nartov, Vladimir V. Kuznetsov

**Affiliations:** 1K.A. Timiryazev Institute of Plant Physiology, Russian Academy of Sciences, Botanicheskaya Street 35, 127276 Moscow, Russia; 2Bryansk State Technological University of Engineering, 3, Stanke Dimitrova St., 241037 Bryansk, Russia

**Keywords:** water stress, mineral nutrition, *Pinus sylvestris*, *Picea abies*

## Abstract

The deterioration of plant mineral nutrition during drought is a significant factor in the negative influence of drought on plant performance. We aimed to study the effects of seasonal and multiyear water shortages on nutrient supply and demand in Scots pine and Norway spruce. We studied pine and spruce trees naturally grown in the Bryansk region (Russia). The dynamics of several nutrients (K, Ca, Mg, P, Fe, Mn, Zn, and Ca) in wood, needles, and bark of current-year twigs and the dynamics of the available pools of these elements at different soil depths were analysed. To assess the physiological consequences of changes in element concentrations, lipid peroxidation products and photosynthetic pigments were measured in the needles. Water shortage increased the wood concentrations of all elements except for Mn. In pine, this increase was mainly due to seasonal water deficit, whereas in spruce, multiyear differences in water supply were more important. This increased availability of nutrients was not observed in soil-based analyses. In needles, quite similar patterns of changes were found between species, with Mg increasing almost twofold and Fe and Mn decreasing under water shortage, whereas the remainder of the elements did not change much under differing water supplies. Neither the concentrations of photosynthetic pigments nor the contents of lipid peroxidation products correlated with element dynamics in needles. In summary, water shortage increased the availability of all elements except Mn for the plant; however, needle element contents were regulated independently of element availability for plants.

## 1. Introduction

In recent years, the mutual effects of plant mineral nutrition and plant performance under drought have begun to receive increased attention [1,2,3].

Water deficit can influence plant mineral nutrition on multiple levels. Drought influences all three ways of nutrient acquisition by roots, namely, diffusion, mass flow, and root interception [4]. Transpiration decrease can hamper nutrient transport in the xylem; however, quite low transpiration rates are found to be sufficient to sustain nutrient transport [5]. Finally, drought-induced growth inhibition reduces plant demand for nutrients; however, increased amounts of some elements can be required for processes such as osmoregulation and turgor maintenance. [6]. As a result, the effects of water stress on plant mineral nutrition are highly complex, being species-, nutrient-, and context-dependent [3,7]; to date, our ability to predict these effects is quite limited.

Multiple recent studies have focused on the impacts of drought on plant mineral nutrition, with controversial results. Drought-stressed seedlings of several drought-sensitive pine species were proposed to face an “isohydric trap”—a feedback cycle of nutrient deficit in leaves that exacerbates the negative drought effect on the plant’s performance [8]. The use of deeper water sources during drought is often correlated with lower element contents in the wood [9] and leaves [10]. However, manipulative experiments demonstrated that increased temperature and not water deficit per se can be responsible for a deterioration of leaf mineral nutrition and for corresponding photosynthetic disturbances in drought-stressed plants [7,11]. Compared with healthy naturally grown trees, declining trees can have access to similar soil nutrient supply rates and can have even higher wood nutrient concentrations [12,13]. As the wood composition at least partially reflects the chemical composition of the tree environment at the time of wood formation [14,15,16], the decline in trees was obviously not due to the reduced nutrient supply. In summary, the current evidence questions whether drought deteriorates tree mineral nutrition and whether such deterioration hinders plant performance and survival during drought.

One possible reason for such discrepancies is that studies often address element contents either in leaves, which are the most physiologically active aboveground plant organs and have the highest demand for mineral nutrients, or in wood in the case of dendrochronological studies. However, nutrient dynamics can differ substantially between organs, and the nutrient contents in leaves are likely more strictly regulated compared with other plant organs [5]. It is clearly important to understand whether the observed drought-induced changes in nutrient contents are the inevitable consequences of water stress or the result of tightly regulated processes of nutrient transport. For this purpose, simultaneous analysis of nutrient dynamics in soil and different plant organs under water shortage is needed. Time is an additional complicating factor as mineral nutrition can be influenced differently by short-term water deficit and long-term impairment of water availability [6,17]. Therefore, we aimed to study the effects of long-term and short-term water deficiency on the mineral contents of wood, needles, and bark of current-year twigs of Scots pine (*Pinus sylvestris* L.) and Norway spruce (*Picea abies* (L.) H. Karst) as well as the effects of water deficiency on the available pools of mineral elements in soils under the plants. We studied trees grown under natural conditions [18] as the influence of water shortage on mineral nutrition can differ substantially between short-term experimental drought and long-term natural differences in water availability [6].

We aimed to study the following issues:-the influence of water supply on the dynamics of available pools of mineral elements in soils and on their contents in plants;-the interrelations of nutrient dynamics between wood, needles, and bark under short- and long-term water shortages;-physiological effects of observed changes in element concentrations in needles.

## 2. Results

### 2.1. pH and Nutrient Dynamics in Soil

The soil water availability at the experimental sites used in the study was characterized in detail in our previous study [18]. In short, based on the dynamics of soil water content relative to field capacity and site live ground cover, Site III (arid for pine) was more arid than Site II (normal for pine and arid for spruce), whereas Site I (normal for spruce) was moister than Site II. The soil water content relative to field capacity decreased substantially across the soil profile at all three sites from the beginning (3 August) to the end (24 August) of the rainless period [18].

When comparing Site II (normal for pine) and Site III (arid for pine), the pH level was higher at depths of 5–15 and 110–120 cm at the more arid Site III (Figure 1A, Appendix A). However, the pH decreased from the beginning by the end of the rainless period across the soil profile. Among all elements, the available K content demonstrated the most prominent changes under different water supplies, being higher at the more arid Site III than at Site II and increasing by the end of the rainless period across the soil profile (Figure 1C). The available Mn content was higher at the arid Site III across the profile except at a depth of 60–70 cm. In contrast, the available P content was lower at the arid site across the profile except at depths of 5–15 cm. For other elements, no consistent dependence of availability on either site or sampling date was found. For Cu, the measurements were not quite reliable (see Materials and Methods) and therefore are not presented here.

When comparing Site I (normal for spruce) and Site II (arid for spruce), the pH level was higher at the more arid Site II than at Site I across the soil profile, but did not change during the rainless period (Figure 1B, Appendix A). Among the elements, the contents of available K and Mn were lower at the arid Site II across the profile except at depths of 5–15 cm, whereas the available Zn content was lower at depths of 60–70 to 110–120 cm (Figure 1D). The available Mg was lower at the arid site at depths of 25–35 and 60–70 cm. For other elements, no consistent dependence of nutrient availability on either site or sampling date was found.

### 2.2. Nutrient Dynamics in Plant Organs

In the wood of current-year pine twigs, a significant increase during the rainless period was observed for 6 out of 8 elements (K, Ca, Mg, P, Zn, and Cu, Figure 2A, Appendix A). The Mg concentrations were also positively influenced by higher site aridity, with no interaction between site aridity and rainless periods. For the Fe content in pine wood, a significant interaction between site and sampling date was observed, with a clear tendency to increase during the rainless period. The concentrations of 7 elements (K, Ca, Mg, P, Fe, Zn, and Cu) in pine wood demonstrated significant positive correlations with each other, except for Fe-P, Fe-K, and Cu-Ca pairs, where the correlation was nonsignificant (Table 1). In spruce wood, a significant positive effect of higher site aridity was observed for five out of eight elements (K, Ca, P, Fe, and Zn, Figure 2B). The Ca and Zn concentrations also increased by the end of the rainless period, with no interaction between site aridity and rainless period for these two elements. The wood Mg and Cu concentrations were positively influenced by the interaction between site aridity and sampling date, and they were the highest at arid Site II by the end of the rainless period. As in pine wood, the concentrations of seven elements (K, Ca, Mg, P, Fe, Zn, and Cu) in spruce wood demonstrated significant positive correlations with each other (Table 1). A distinct accumulation pattern was observed for Mn in both species as the accumulation of this element in wood was negatively influenced by the rainless period in pine and by higher plot aridity in spruce. In pine wood, the Mn content was negatively correlated with the contents of Mg, Fe, and Ca, whereas in spruce wood, no significant correlations between the contents of Mn and other elements were found (Table 1).

Changes in element concentrations in needles and bark were generally less pronounced than those in wood (Figure 2C–F). The needle and bark Mg contents increased during the rainless period in both species, whereas Mn and Fe decreased in response to water shortage, except for spruce bark, where no significant changes in Fe content were observed. The dynamics of Mg and Mn in needles and bark were similar to those in wood, and significant positive correlations between the contents in wood, needles, and bark were observed for each of these two elements (Table 2). The Cu concentration in spruce bark increased in response to the rainless period, whereas in needles, a complex interaction between site aridity and sampling date was observed. The contents of K, Ca, P, and Zn in the needles of both species, as well as the Cu contents in pine needles, were unresponsive to water shortage despite the almost uniform increase in the contents of these elements in the wood either by the end of the rainless period (in pine) or in the arid plot (in spruce).

### 2.3. Concentrations of Lipid Peroxidation Products and Photosynthetic Pigments

Both Fe and Mg are essential metals for chlorophyll biosynthesis. At the same time, Fe and Mn are transition metals able to induce oxidative damage; therefore, changes in concentrations of these elements can potentially result in alterations of plant oxidative balance. Therefore, to assess the physiological importance of observed changes in Mg, Fe, and Mn concentrations in needles, we measured the concentrations of photosynthetic pigments and lipid peroxidation products in needles.

In pine needles, MDA content increased during the rainless period, but no significant changes were observed for 4-HNE (Figure 3A, Appendix A). A weak but significant negative correlation (*r* = −0.47) was observed between Mn content and MDA content in pine needles (Table 3). In spruce needles, differently directed changes were observed for MDA concentrations, and no significant changes were observed for 4-HNE (Figure 3B). No significant correlations between the contents of peroxidation products and either Mn or Fe concentrations were observed in spruce needles (Table 3).

In pine, the Chl *a* and Chl *b* concentrations decreased during the rainless period, and the Chl *a*/Chl *b* ratio was lower in the arid plot (Figure 3C, Appendix A). The carotenoid concentrations were not influenced either by plot or by the rainless period, which resulted in an increased Car/Chls ratio by the end of the rainless period (Figure 3C). The chlorophyll concentrations were not significantly correlated with either needle Mg or Fe content; the only significant correlation was between Mg content and Car/Chls ratio (*r* = 0.36) (Table 4). In spruce, the concentrations of Chl *a*, Chl *b*, and carotenoids were lower in the arid plot (Figure 3D). Water stress did not influence the Chl a/Chl *b* ratio, and the Car/Chls ratio changed in different directions, depending on the interaction between plot aridity and sampling date (Figure 3D). The needle Mg concentrations in spruce demonstrated a positive correlation with the Car/Chls ratio (*r* = 0.56), whereas a weak negative correlation was observed between the Car/Chls ratio and Fe content (*r* = −0.44) (Table 4).

## 3. Discussion

Plant-based analysis of nutrient contents in wood can capture multiple drought effects on element availability, absorption, and consumption and is therefore assumed to adequately reflect the changes in the nutrient balance of trees under different environmental conditions [13,14]. Although it is proposed that the impact of drought on nutrient uptake is species- and nutrient-dependent [3], we found that the majority of elements exhibited rather similar wood accumulation patterns within each species. In pine, wood concentrations of seven out of eight elements (K, Ca, Mg, P, Fe, Zn, and Cu) were positively influenced by the rainless period (Figure 2A), whereas in spruce, higher concentrations of five elements (K, Ca, P, Fe, and Zn) were observed in the wood of plants grown in the arid plot, with two additional elements (Mg and Cu) demonstrating a positive interaction between site aridity and rainless periods (Figure 2B). When analysing the dynamics of leaf mass per area and osmotic pressure in the same pine and spruce plants [18], we observed that pine plants were impacted mainly by short-term water shortage during the rainless period, whereas in spruce plants, higher plot aridity was the main cause of decreased water supply. Therefore, we suppose that the decreased soil water supply positively influenced the availability of the majority of nutrients for plants and therefore increased nutrient concentrations in the current-year wood.

The wood manganese concentrations were lower by the end of the rainless period in pine and in the arid plot in spruce, thus demonstrating an accumulation pattern opposite to that of the majority of other elements in each species. It is well known that changes in Mn accumulation are characteristic of different disturbances in trees [see, e.g., [10,13]]. Drought-stressed beech seedlings had decreased Mn concentrations in roots and stems but not in leaves [5], and a large analysis of leaf element concentrations in almost 2000 plant species in China demonstrated that leaf Mn concentrations increased with increasing site mean annual precipitation, in contrast to other essential elements, which increased from more humid to more arid sites [19]. Our study confirms the opposite influence of water supply on Mn accumulation compared with other nutrients studied. However, although such a distinct pattern of Mn accumulation has often been found, neither the causes nor physiological consequences of this accumulation pattern are currently known.

Compared with the almost uniform effect of decreased water supply on element concentrations in the wood, relatively few changes in the content of available elements were found in the soil analysis. In some cases, these changes coincided with the element dynamics in the wood; for example, the K accumulation in pine wood (Figure 2A) coincided with the increased available soil K by the end of the rainless period (Figure 1C), and the decreased Mn content in the wood of spruce grown on the arid site (Figure 2B) coincided with lower available soil Mn on this site (Figure 1D). In other cases, the opposite dynamics were observed. The potassium concentrations in spruce wood (Figure 2B) were influenced positively by site aridity despite the decrease in available soil K in the arid plot (Figure 1D). The Mg concentrations in spruce wood were positively influenced by the reduced water supply (Figure 2B) despite the decreased soil Mg availability in the arid plot (Figure 1D). Finally, in some cases, the changes in soil nutrient availability did not influence wood element concentrations; for example, neither P nor Mn were influenced by plot aridity in pine wood (Figure 2A) despite the decreased available P and increased available Mn in the arid plot (Figure 1C). Therefore, there was no consistent link between the contents of available elements in the soil and their accumulation in the current-year wood. It is well known that soil testing methods only characterize some of the factors that influence nutrient supply to the roots of plants [20], and the assessment of the available fraction of nutrients in the soil, which is quite a useful method for studying, e.g., soil pollution effects [21], is probably of limited value when the effects of water deficit are analysed. Our data indicate that it is crucial to perform not only soil-based but also wood-based analyses of element concentrations to adequately assess the availability of nutrients for plants during water shortage.

The leaf is the major sink for mineral nutrients, and both the deficiency and excessive content of mineral elements can be detrimental to physiological processes in leaf tissues. As a result, leaf nutrient concentrations can be maintained strictly even under substantial drought-induced nutrient shortages [5], being only weakly related to nutrient concentrations in the wood or in the soil [14]. To make physiologically meaningful conclusions on changes in the nutrient status of needles under water shortage, the dynamics of a given element in needles and wood should be compared. Additionally, the mere fact of an increase or decrease in nutrient concentration does not indicate the physiological excess or deficiency of this nutrient; therefore, the observed changes should be compared with known ranges of optimal nutrient concentrations in needles in pine and spruce. Based on these comparisons, we can specify three groups of elements from our data.

The first group of elements includes Mg in both species. The concentration of Mg increased in wood, needles, and bark in a coordinated way, mainly in response to the rainless period (Figure 2, Table 2). The needle Mg concentrations were lower than the optimum (0.8 mg/g DW) at the beginning of the rainless period in both species, increasing to the optimal range (>0.8 mg/g DW) by the end of the rainless period (Figure 4A). It can be proposed that increased Mg availability under water shortage, indicated by increased Mg concentrations in the current-year wood, was utilized by plants to bring needle Mg concentrations to the optimal range. However, in both species, no significant positive correlations were observed between the contents of chlorophylls *a* and *b* and the Mg content in needles. Mg is required for chlorophyll biosynthesis, and decreased Chl content as well as an increased Car/Chls ratio are observed in Mg-deficient plants, leading to leaf chlorosis [20]. Therefore, the absence of positive correlations between needle Mg and chlorophyll contents argues against the proposition that control pine and spruce plants suffered from Mg deficiency, which was remedied under conditions of water shortage during the rainless period.

The second group of elements was the most represented and included K, Ca, P, and Zn in both species and Cu in pine. The concentrations of these elements were quite stable in needles and did not change much in bark despite the substantial increase in wood under conditions of water shortage. Excessive nutrients can be actively transported to the other plant parts by the phloem; by this transport, it is possible to maintain the total needle concentration of elements on an unchanged level despite their increased supply from the xylem. However, calcium has rather low phloem mobility [22] and was still maintained at the unchanged level in needles and bark despite the substantial increase in the wood of water-limited trees. Therefore, the mechanisms by which the needle concentrations of the aforementioned elements are maintained at a stable level remain to be determined.

The third group of elements includes Fe and Mn, which decreased prominently and uniformly in needles of both species under water shortage. The Fe concentrations could be considered optimal in spruce and optimal or suboptimal in pine (Figure 4). Together with no significant correlations between Fe content and Chl content, the results speak to the absence of physiological Fe deficiency in water-limited trees despite the decreased needle Fe content. The decrease in Fe content in needles of water-stressed plants occurred despite the generally positive influence of water shortage on wood Fe content in both species (Figure 2A,B). The Mn concentrations were within the optimal range in Scots pine, which is known to require a high Mn supply compared with other pine species [23,24], whereas a clear Mn excess was observed in spruce needles [20]. Although decreased wood Mn concentrations indicated decreased Mn supply under water shortage, this was unlikely to decrease the needle Mn concentrations, as the needle growth was already finished by the beginning of August; therefore, no “growth dilution” was possible. We claim that such a profound decrease indicates the active remobilization of both Fe and Mn from the needles under conditions of water shortage. This is quite surprising, as Fe is considered to have intermediate phloem mobility while Mn has low phloem mobility [20]. In *Picea abies*, the concentration ratios of Fe and Mn between young and old needles were very close to their phloem mobility ratios, which implies that Fe and Mn remobilization is indeed restricted by their phloem mobility [25]. On the other hand, a detailed analysis of Mn transport in *Pseudotsuga menziesii* demonstrated that Mn can be transported by the phloem, albeit at a quite slow rate, and that this transport is important for maintaining needle Mn concentrations under increased Mn uptake by roots [26].

Fe and Mn were significantly correlated in both pine and spruce needles (Table 1). One possible reason for such similar changes could be that both metals are remobilized by the same set of transporters, and therefore their transport occurs in the same manner. For example, NRAMP3, NRAMP4, and several MTP transporters have broad substrate specificity and are involved in both Fe and Mn transport within cells [32,33]. Activation of these transporters can lead to concurrent Fe and Mn remobilization from the cells; however, we failed to identify the close homologues of genes encoding these transporters in Pinaceae plants (results not shown) and were unable to test this hypothesis. Another possible reason for such co-regulation is that both Fe and Mn are transition metals with varying oxidative states and can participate in Fenton reactions [34]. Under water stress, the generation of superoxide radicals and H_2_O_2_ in the photosynthesis electron transport chain is often increased, leading to Fe- and Mn-catalysed generation of highly reactive hydroxyl radicals [35]. Therefore, the decrease in “surplus” manganese and iron in photosynthetic tissues can be a measure to decrease the risk of oxidative damage under conditions of water shortage. However, we observed no negative correlations between the contents of lipid peroxidation products and the contents of either Fe or Mn in needles. Therefore, the reasons for the observed similarity between the dynamics of Fe and Mn in needles under water stress are unknown.

## 4. Materials and Methods

### 4.1. Selection of Experimental Sites

Investigations were performed in native undisturbed pine and spruce stands of productivity class I in the Bryansk region (Russia). Based on the annual growth data, 3 experimental sites were selected for this study [18]:-Site I (53 10.923, 34 34.334). Mean age of adult trees, 70 years; productivity class, I; relative density, 0.8; stand composition, 10 spruce; forest site type, B_3_; forest type, bilberry-spruce forest; and soil type, slightly soddy sandy mesopodzol with a ferruginous illuvial horizon on fluvioglacial sand. Based on the living ground cover species, this site is mesohydrophilous [36], and it was therefore considered a site with a normal water regime for spruce (designated N).-Site II (53 11.525, 34 34.760). Mean age of adult trees, 70 years; productivity class, I; relative density, 0.9; stand composition, 8.7 pine, 1.3 spruce + birch; forest site type, A_2_; forest type, cowberry pine forest; and soil type, medium soddy sandy mesopodzol on fluvioglacial sand, with quartz-glauconitic sand containing phosphorites underneath. Based on the living ground cover species, this site is mesophilic [36], and it was therefore considered arid (A) for spruce and normal (N) for pine.-Site III (53 13.785, 34 35.352). Mean age of adult trees, 60 years; productivity class, I; relative density, 0.5; stand composition, 10 pine; forest site type, A_1_; forest type, cowberry-pine forest; and soil type, slightly soddy sandy cryptopodzol on fluvioglacial sands. Based on the living ground cover species, this site is mesoxerophilous [36], and it was therefore considered arid (A) for pine.

In total, for each species, one normal (N) and one arid (A) site were studied; see [18] for a more detailed description of site selection.

### 4.2. Collection of Plant and Soil Samples

Plant and soil samples were collected in 2018. The year 2018 was relatively hot and dry compared with both climatological reference normals (1961–1990 and 1981–2010), and August was the driest month of the vegetation season, with a rainless period occurring from 2 August to 29 August (see [18] for detailed description). Plant and soil sampling was performed at the beginning and end of the rainless period. The first sampling was performed on 3 August at the normal and arid sites, hereafter designated N3 and A3, respectively. The second sampling was performed on 24 August at the normal and arid sites, hereafter designated N24 and A24, respectively.

Current-year twigs were collected from the lateral shoots of the undergrowth of pine (7–15 years) and spruce (10–25 years) around the perimeter of the tree crown. For sampling, 8 plants of each species were chosen in the normal and arid sites, with each plant taken as a biological replicate. Twigs were excised, wrapped in plastic bags, placed on ice, transported to the laboratory within 12 h, and used for the determination of nutrient contents in current-year wood, needles, and bark.

Soil samples were collected from depths of 5–15 cm, 25–35 cm, and 60–70 cm on 3 August. During sample collection, the roots of pine and spruce undergrowth were found to have penetrated the soil profile to depths lower than 60–70 cm. Therefore, during soil sampling on 24 August, samples were also collected from depths of 110–120 cm. For each site and depth, 4 soil samples were taken as biological replicates. The soil samples were placed in hermetically sealed plastic containers and transported to the laboratory for the determination of soil pH and available pools of mineral nutrients. The collected soil samples were air-dried and then sieved to obtain fractions <2 mm in size. Before the analysis, a small portion of each soil sample was oven-dried at 105 °C to correct the dry weight results [37].

### 4.3. Analysis of Soil pH and Available Pools of Mineral Nutrients

The soil pH was measured in a 1:5 (*v*/*v*) suspension of soil in a 0.01 M CaCl_2_ solution (pH in CaCl_2_) [38].

The available pools of K, Mg, Fe, Mn, Zn, and Cu were determined in duplicate by 1:10 (*w*/*v*) extractions with 0.01 M CaCl_2_, in which 2.5 g of soil and 25 mL of solution were used [37]. The available pools of Ca and P were determined in duplicate via 1:5 (*w*/*v*) extractions with 1 M CH_3_COONH_4_, in which 1 g of soil and 5 mL of solution were used [39]. After being shaken for 2 h at room temperature, the samples were filtered (0.45 µm filter), after which the extraction fluids were used for elemental analysis. All elements except P were determined by atomic absorption spectroscopy [21]. Note that the content of bioavailable Cu was lower than the detection limit; therefore, the data obtained for the available Cu content in the soil are not quite reliable and not given. Phosphorous was determined using a molybdenum blue reaction [40]. Then, 0.5 ml of the diluted sample was added to the reaction mixture, which contained 100 mM H_2_SO_4_, 0.85 mM (NH_4_)_6_Mo_7_O_24_, and 43 mM ascorbic acid. After colour development incubation for 3.5 h at 25 °C, the optical density was measured at 825 nm. To construct the calibration curve, KH_2_PO_4_ was used.

### 4.4. Analysis of the Elemental Composition of Wood, Needles, and Bark of Current-Year Twigs

The collected twigs were washed in 10 mM aqueous solutions of EDTA·Na_2_; separated into wood, needles, and bark; and dried to constant weight at 80 °C for three days. The samples were then digested in solutions of concentrated HNO_3_ and HClO_4_ (2:1 (*v*/*v*)) [21], after which the elemental content was determined as described earlier.

### 4.5. Evaluating the Level of Lipid Peroxidation in the Needles

The contents of malondialdehyde (MDA) and 4-hydroxy-2-nonenal (4-HNE) were determined spectrophotometrically, with maximum optical absorption at 586 nm, by measuring the product that formed during the reaction using the selective reagent 1-methyl-2-phenylindole (Aldrich, CAS Number 3558-24-5) in accordance with [41]. To construct the calibration curve, 1,1,3,3-tetraethoxypropane was used.

### 4.6. Determination of Photosynthetic Pigments in the Needles

The chlorophyll *a*, *b*, and carotenoid contents were determined using the Lichtenthaler method [42]. The samples were triturated with 80% acetone in the dark. The absorbance of the samples was measured with a Genesys 10 UV–Vis spectrophotometer (Thermo Fisher Scientific, Waltham, MA, USA) at wavelengths of 470, 646, and 663 nm. The content of the photosynthetic pigments was calculated as described in [42].

### 4.7. Statistical Analysis

Each tree was treated as a biological replicate; therefore, there were 8 biological replicates for the plant-based analyses of nutrient contents in needles, bark, and wood and of the contents of pigments and lipid peroxidation products in needles. For soil analyses, there were 4 biological replicates. The data were statistically analysed using SigmaPlot 12.3 (Systat Software, United States). To separate the effects of site and sampling date, two-way ANOVA was used (*p* < 0.05). To calculate the strength and significance of correlations between variables, Pearson’s correlation was used. The Pearson’s correlation coefficients presented in the text are significant at *p* < 0.05. The values presented in the figures are the arithmetic means, and the values presented in the tables are arithmetic means ± standard errors.

## 5. Conclusions

Based on the wood element concentrations, our results argue in favour of increased rather than decreased bioavailability of the majority of nutrients (except Mn) under water shortage. In pine, this increase was mainly due to seasonal water deficit after the rainless period, whereas in spruce, multiyear differences in water supply between normal and arid plots were more important. This increase was not revealed by soil analyses, indicating the limited ability of soil-based methods to assess changes in nutrient availability for trees under water shortage. We demonstrated that the influence of water shortage on tree nutrient status can differ substantially depending on the analysed organs or tissues, with the nutrient status of needles being the least affected and that of wood being the most affected by water shortage. This indicates the necessity of simultaneous analysis of wood and leaf element concentrations to better understand the influence of water shortage on tree nutrient supply, in contrast to the most common approach, where physiological conclusions are made based on either the leaf or wood element composition. Only three elements, namely, Mg, Mn, and Fe, changed consistently in the needles of both species in response to the decreased water supply. However, no strong correlations of these elements with the contents of photosynthetic pigments and lipid peroxidation products were found, thus suggesting that these changes in nutrient concentrations neither maintained the functioning of the photosynthetic apparatus nor decreased the risk of oxidative damage under water stress. Therefore, the physiological importance of strict maintenance of Mg, Mn, and Fe concentrations in needles remains to be identified.

## Figures and Tables

**Figure 1 plants-11-02652-f001:**
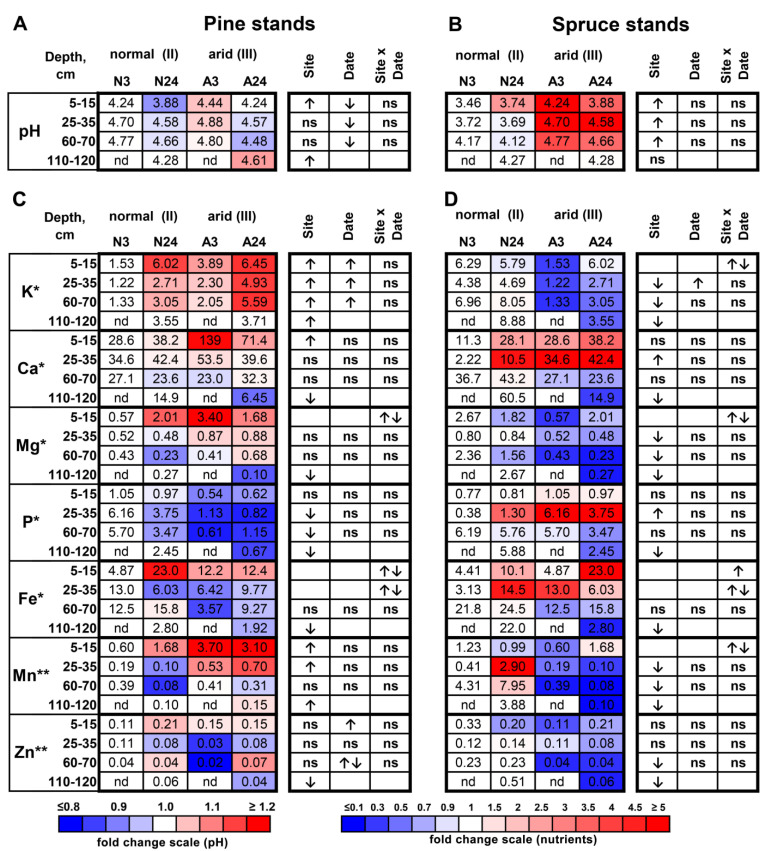
pH (**A**,**B**) and contents of nutrients (**C**,**D**) at 4 depths (I—5–15 cm, II—25–35 cm, III—60–70 cm, and IV—110–120 cm) for pine-inhabited Site III (arid) and Site II (normal) (**A**,**C**) and for spruce-inhabited Site II (arid) and Site I (normal) (**B**,**D**). The contents of the elements on 3 August in the normal plot were taken as 1.0 (white), the relative increase is indicated by red, and the relative decrease is indicated by blue; numbers indicate mean element contents in mg/kg DW (* and **). The significance of the site, sampling date, and site × date interaction were calculated using two-way ANOVA (*p* < 0.05), with an upwards arrow indicating positive influence, a downwards arrow indicating negative influence, double arrows indicating differently directed influence, and “ns” indicating no significant influence.

**Figure 2 plants-11-02652-f002:**
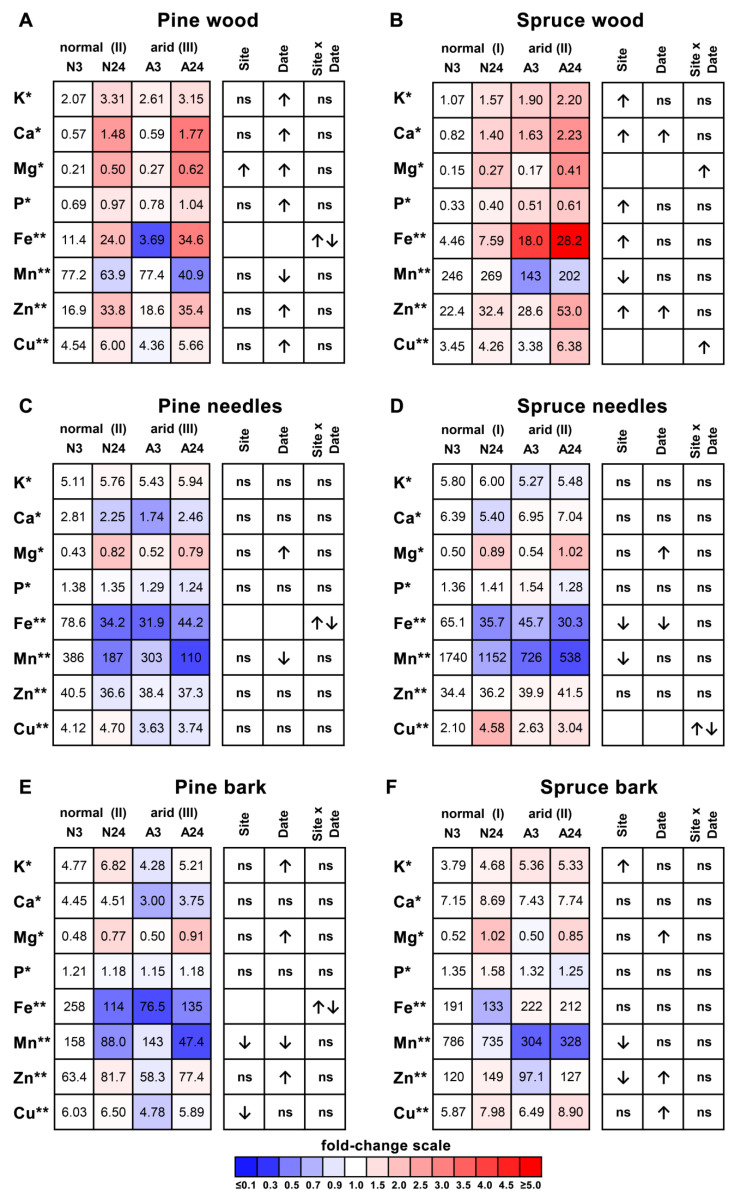
Nutrient contents in the wood (**A**,**B**), needles (**C**,**D**), and bark (**E**,**F**) of pine (**A**,**C**,**E**) and spruce (**B**,**D**,**F**). The nutrient contents on 3 August in the normal plot were taken as 1.0 (white), the relative increase is indicated by red, and the relative decrease is indicated by blue; numbers indicate mean element contents in mg/g DW (*) or µg/g DW (**). The significance of the site, sampling date, and site × date interaction were calculated using two-way ANOVA (*p* < 0.05), with an upwards arrow indicating positive influence, a downwards arrow indicating negative influence, double arrows indicating differently directed influence, and “ns” indicating no significant influence.

**Figure 3 plants-11-02652-f003:**
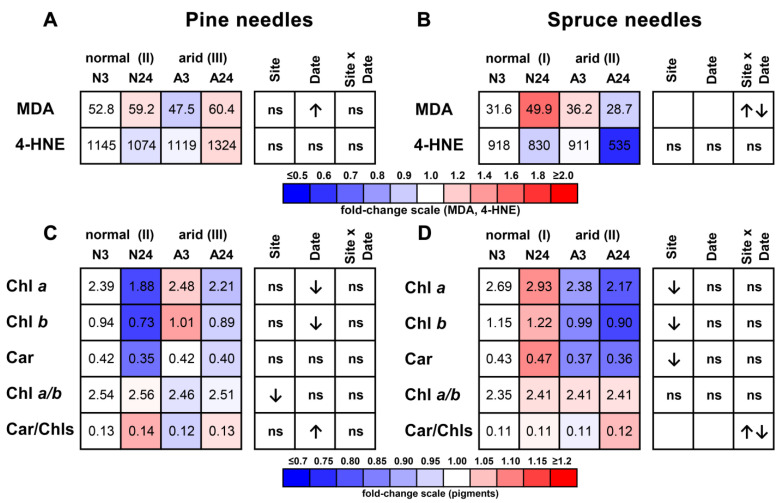
Contents of lipid peroxidation products (**A**,**B**) and photosynthetic pigments (**C**,**D**) in pine (**A**,**C**) and spruce (**B**,**D**) needles. The contents of the compounds on 3 August in the normal plot were taken as 1.0 (white), the relative increase is indicated by red, and the relative decrease is indicated by blue; numbers indicate mean compound contents (nmol/g DW for MDA and 4-HNE, and mg/g DW for chlorophylls *a* and *b* (Chl *a*, Chl *b*) and carotenoids (Car)). The significance of the site, sampling date, and site × date interaction were calculated using two-way ANOVA (*p* < 0.05), with an upwards arrow indicating positive influence, a downwards arrow indicating negative influence, double arrows indicating differently directed influence, and “ns” indicating no significant influence.

**Figure 4 plants-11-02652-f004:**
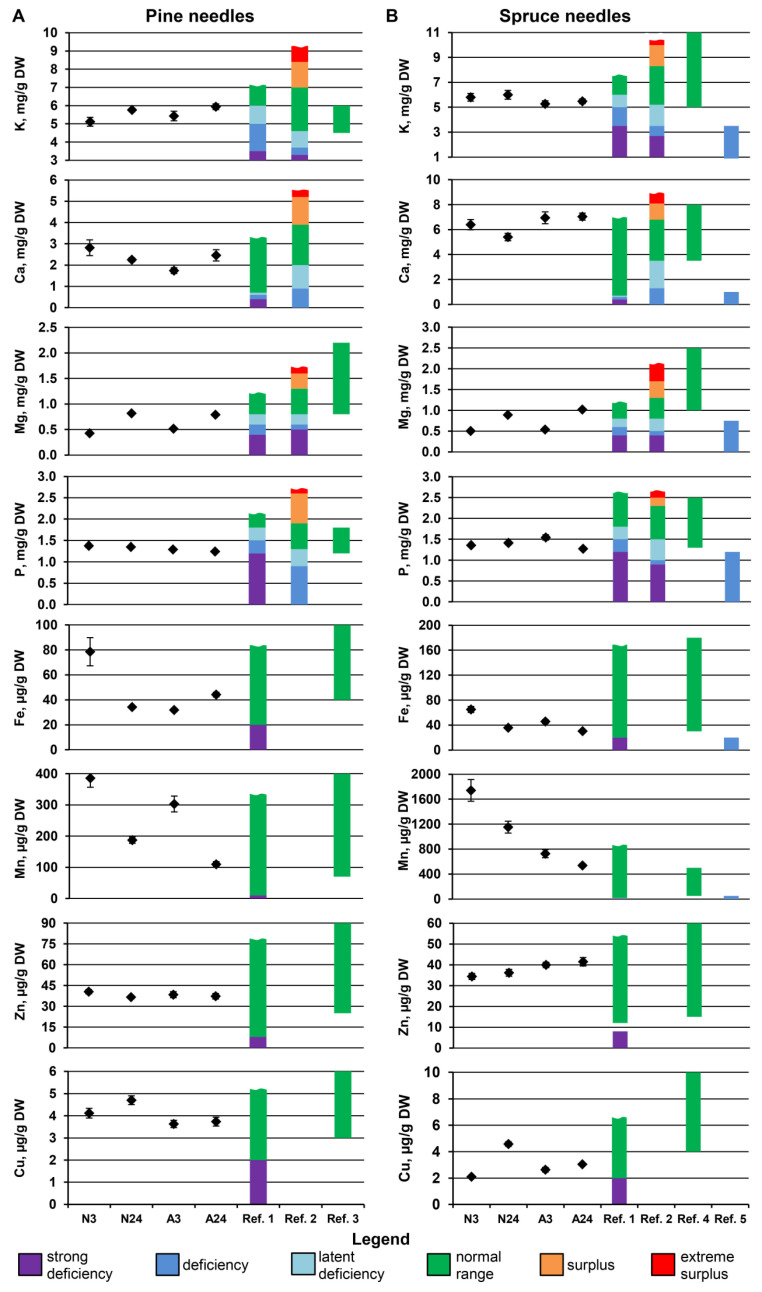
Comparison of the changes in element concentrations in pine (**A**) and spruce (**B**) needles with a known range of optimal nutrient concentrations. The mean values ± SEs are given (n = 8). Ref. 1—[27]; Ref. 2—[28]; Ref. 3—[29]; Ref. 4—[20], (for Fe—[30]); Ref. 5—[31].

**Table 1 plants-11-02652-t001:** Pearson’s correlation coefficients (*r*) between element contents in plant organs.

	Ca	Mg	P	Fe	Mn	Zn	Cu	Ca	Mg	P	Fe	Mn	Zn	Cu
	Pine wood							Spruce wood						
K	0.36 *	0.63 ***	0.89 ***	0.30	0.05	0.66 ***	0.40 *	0.68 ***	0.66 ***	0.88 ***	0.71 ***	0.04	0.53 **	0.53 **
Ca		0.69 ***	0.48 **	0.60 ***	−0.46 **	0.53 **	0.34		0.72 ***	0.81 ***	0.76 ***	0.07	0.90 ***	0.47 **
Mg			0.71 ***	0.72 ***	−0.53 **	0.82 ***	0.61 ***			0.73 ***	0.63 ***	0.24	0.75 ***	0.81 ***
P				0.29	−0.07	0.65 ***	0.45 **				0.73 ***	0.07	0.68 ***	0.58 ***
Fe					−0.54 **	0.69 ***	0.43 *					0.01	0.73 ***	0.60 ***
Mn						−0.16	−0.19						0.21	0.21
Zn							0.47 **							0.52 **
	Pine needles							Spruce needles						
K	−0.50 **	0.27	0.48 **	−0.53 **	−0.43 *	−0.38 *	0.50 **	−0.39 *	0.14	0.42 *	−0.26	0.11	0.28	0.03
Ca		−0.02	−0.41 *	0.80 ***	0.45 **	0.55 ***	−0.40 **		0.07	−0.42 *	0.26	0.17	0.36 *	−0.38 *
Mg			−0.05	−0.31	−0.51 **	0.04	0.19			−0.08	−0.60 ***	−0.18	0.33	0.54 **
P				−0.27	−0.11	−0.28	0.73 ***				−0.20	0.00	0.10	0.30
Fe					0.66 ***	0.40 *	−0.26					0.48 **	−0.18	−0.44 *
Mn						0.55 **	−0.25						0.05	−0.20
Zn							−0.22							−0.12
	Pine bark							Spruce bark						
K	0.05	0.39 *	0.29	−0.31	−0.31	0.37 *	0.48 **	−0.11	0.19	0.34	−0.32	−0.13	−0.26	0.26
Ca		0.30	−0.07	0.67 ***	0.09	0.13	0.21		0.27	0.20	−0.12	0.17	0.59 ***	−0.03
Mg			0.15	0.02	−0.59 ***	0.67 ***	0.48 **			0.44 *	−0.50 **	0.21	0.35 *	0.59 ***
P				−0.23	−0.11	0.24	0.22				−0.69 ***	0.34	0.12	0.11
Fe					0.23	−0.00	0.17					−0.30	−0.14	−0.36 *
Mn						−0.16	−0.15						0.29	−0.08
Zn							0.49 **							0.02

* Pairs with *p* < 0.050; ** pairs with *p* < 0.010; *** pairs with *p* < 0.001.

**Table 2 plants-11-02652-t002:** Pearson’s correlation coefficients (*r*) for each element between plant organs.

Nutrients	Scots Pine			Norway Spruce		
	Wood–Needles	Wood–Bark	Needles–Bark	Wood–Needles	Wood–bark	Needles–Bark
K	0.50 **	0.44 *	0.58 ***	0.34	0.61 ***	0.48 **
Ca	0.31	0.30	0.81 ***	0.06	−0.20	0.08
Mg	0.78 ***	0.73 ***	0.76 ***	0.68 ***	0.53 **	0.60 ***
P	0.29	0.17	0.33	0.18	−0.25	0.22
Fe	0.06	0.17	0.87 ***	−0.48 **	0.40 *	0.17
Mn	0.59 ***	0.71 ***	0.82 ***	0.62 ***	0.69 ***	0.86 ***
Zn	−0.11	0.66 ***	0.15	0.24	0.24	0.04
Cu	0.49 **	0.71 ***	0.49 **	0.15	0.62 ***	0.40 *

* Pairs with *p* < 0.050; ** pairs with *p* < 0.010; *** pairs with *p* < 0.001.

**Table 3 plants-11-02652-t003:** Pearson’s correlation coefficients (*r*) between the contents of certain elements and lipid peroxidation products.

Elements	Pine Needles		Spruce Needles	
	MDA	4-HNE	MDA	4-HNE
Mn	−0.47 **	−0.10	0.09	0.05
Fe	0.02	−0.04	−0.23	0.10

Pairs with *p* < 0.050; ** pairs with *p* < 0.010.

**Table 4 plants-11-02652-t004:** Pearson’s correlation coefficients (*r*) between the contents of certain elements and photosynthetic products.

Elements	Chl *a*	Chl *b*	Car	Chl *a/b*	Car/Chls
Pine needles					
Mg	−0.25	−0.21	−0.14	−0.08	0.36 *
Fe	0.29	0.23	0.34	0.26	0.01
Spruce needles					
Mg	−0.05	−0.06	0.05	0.15	0.56 ***
Fe	0.17	0.20	0.11	−0.29	−0.44 *

* Pairs with *p* < 0.050; *** pairs with *p* < 0.001.

## Data Availability

The datasets generated and/or analysed during the current study are available from the corresponding author upon reasonable request.

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
