# Peer review of "Mineral Nutrition of Naturally Growing Scots Pine and Norway Spruce under Limited Water Supply"

_plants, 2022, doi:10.3390/plants11192652_

Round 1
Reviewer 1 Report
Manuscript title: Mineral nutrition of naturally growing Scots pine and Norway spruce under limited water supply
Manuscript ID: plants-1937818
Journal: Plants
The aim of the current manuscript was to focus on (i) the influence of water supply on dynamics of available pools of mineral elements in soils and on their contents in plants; (ii) the interrelations of nutrient dynamics between wood, needles, and bark under short- and long-term water shortage; (iii) physiological effects of observed changes in element concentrations in needles. The subject is interesting and the manuscript was well-written. However, several points need to be considered before accepting the current version:
- Abstract must be informative including the most important results in values and/or percentages;
- Introduction section is long and need to be reduced focusing on the aim of the study;
- Please remove the subtitles in the discussion section and make it correlated with the findings;
- Try to discuss only your results and remove unnecessary sentences.
Author Response
Answer 1: We changed some formulations in the Abstract. However, we were unable to add the values or percentages in the most cases, since these values differed both between species and between types of stress (long-term vs seasonal) and were therefore unsuitable to be expressed in a short way.
Answer 2: The Introduction section was substantially shortened to sharpen the focus on the objects of the study.
Answers 3-4: The Discussion section was substantially shortened to make it more correlated with the findings; the subtitles were removed.
Reviewer 2 Report
Please find the attached file

Author Response
Answer 1: Thank you, the scientific names were added to the Introduction section.
Answer 2: We apologize for this mistake, it was corrected in the revised version of the Manuscript.
Answer 3: As we mentioned in the Discussion, our previous study demonstrated that pine plants were impacted mainly by short-term water shortage during the rainless period, whereas in spruce plants, higher plot aridity was the main cause of decreased water supply. Therefore, we cannot attribute the observed differences in nutrient accumulation to the differences between species, different accumulation patterns can be attributed to differences between plots as well. We added the mention of differences between species in Abstract and Conclusion.
Answer 4: The required changes were made in Figures.
Answer 5: The required Tables (3 and 4) was added in the revised version of the Manuscript.
Answer 6: The caption for Figure 4 was improved.
Reviewer 3 Report
Dear authors
It has recently been predicted that climate change will create unfavourable conditions for both pines and spruces, which are likely to retreat to northern Europe. They are to be replaced by deciduous species such as oak. In the light of your research, can you support this thesis or not? Does spruce seem more unsuited to changing environmental conditions than pine? Weakened trees are often attacked by bark beetles such as Ips typographus, leading to massive die-offs of entire stands. Could changes in the composition of nutrient elements in the wood favour the insect?
On the other hand, pine trees are attacked by mistletoe on a scale not previously observed, especially with drought causing them to die en masse? Could the recorded concentrations of the elements K, C, Mg, P, Fe, Zn and Cu during periods without rain stimulate mistletoes to grow more intensively?
Conifers appeared to be drought-resistant (e.g. small needle area, wax coverage, etc.), yet it appears that they die off first during climate change? Can a studied nutrient supply mechanism help explain this phenomenon?
Author Response
Answer 1: Thank you for interesting question. Historically (in the mid-XIX century), oak forests occupied about 25-30% of total area covered by forests in the region. However, substantial part of these forests was cut for building and agricultural purposes. Moreover, extreme frosts, dry winters and outbreaks of herbivorous insects resulted in massive oak dieback during the XX century. As a result, oak is not a popular species for reforestation in the region. In addition, oak requires quite rich soils and does not perform well on podzol sandy soils, on which the studied pine and spruce stands were grown. Therefore, the substitution of coniferous species with oaks is unlikely in the region in near future. However, these processes were no within the scope of our research interested and were therefore not discussed in the Manuscript.
Answer 2: It is highly likely since spruce is more water-demanding species with more shallow root system compared to pine. The studied region is on the south edge of spruce distribution area. Also, spruce distribution area moves northward during the last quarter of a century.
Answer 3: Based on our results, we cannot reasonably answer this question. Ips typographus consumes phloem tissues and sapwood. We studied total wood of 1-year shoots, and therefore we are not aware of how the elements are distributed between outer and deeper xylem layers. The separate study is required to answer this question.
Answer 4: We cannot reasonably answer this question. Mistletoe is not found in the region neither on pines nor on other plant species due to low winter temperatures. There are no evidences of mistletoe invasion cases after the drought periods in studied region.
Answer 5: It is unlikely that preferential die-off of coniferous species can be attributed to deterioration of mineral nutrition, including the phenomenons found in the current study. The likely reason for higher susceptibility of coniferous species compare to angiosperms is their lower ability to recover from water deficit and inability to resprout, which makes them more susceptible to multi-year deterioration of stem hydraulic transport compared to angiosperms. Also, the angiosperms are better suit to drought-induced leaf shedding which helps to minimize the residual water loss during severe drought.